# RECOVERING KNOWLEDGE BY HARDENING LANGUAGE MODELS

## ABSTRACT

Recent neural language models show impressive capabilities on a wide range of tasks. However, it is not fully understood how the knowledge of the language is encoded in these models. In this work, we focus on the simplest case of languages, regular languages, and study language models trained on strings matching certain regular expressions. We propose a method, dubbed LaMFA, to recover the full knowledge of the regular language model by *hardening* it into a finite automaton. Such hardening is conducted by empirically partition the latent space of language models into finite states, and then recover a deterministic finite automaton by the estimated transition probabilities between these states. Through experiments on regular languages of varying complexity, we demonstrate that LaMFA can effectively extract DFA that consistently replicate the performance of the original language model. Notably, the extracted DFAs exhibit enhanced generalization capabilities, achieving 100% accuracy even in out-of-distribution scenarios

## 1 INTRODUCTION

Recent progress on large language models (Brown et al., 2020; Chowdhery et al., 2022; OpenAI, 2023) has shown impressive capabilities of neural networks on a remarkably wide range of tasks such as chatbot (OpenAI, 2023), code generation (Chen et al., 2021), math word problem solving (Lewkowycz et al., 2022; Zheng et al., 2023; Yu et al., 2023), theorem proving (Polu & Sutskever, 2020; Jiang et al., 2023; Wang et al., 2023b;a) and even tasks on other modalities such as image classification (Dosovitskiy et al., 2021), text-to-image generation (Koh et al., 2024), VQA (OpenAI, 2023). Some postulate that certain large language models such as GPT-4 have made an important step towards Artificial General Intelligence (AGI) (Bubeck et al., 2023).

Impressive as their achievements are, the idea behind these large language models is strikingly simple. As all languages (and further all sorts of information) consist of sequences of tokens (characters, bits, etc) $x_i$, it all boils down to model the decomposed joint distribution

$$p(x_1, x_2, ..., x_T) = \prod_{t=1}^{T} p(x_i|x_{<t}) \tag{1}$$

for a given $T \in \mathbb{N}$. The key to the success of large language models lies in their ability to compress information by learning this objective (Schmidhuber & Heil, 1996; Deletang et al., 2024). By training on vast corpora of text, language models effectively learn to compress the statistical regularities and patterns inherent in language. This compression process leads to the strong generalization performance observed in state-of-the-art LLMs (Deletang et al., 2024).

Nonetheless, the nature of the compressed knowledge encoded within neural language models remains largely opaque. Though efforts have been made by probing factual knowledge (Jiang et al., 2020) or syntax concepts (Shi et al., 2016; Tenney et al., 2019), the internal representations and decision-making processes of natural language models remain unclear. This lack of interpretability poses significant challenges, particularly when it comes to addressing issues such as hallucinations (Brown et al., 2020; Zhang et al., 2023), where models generate false or nonsensical information with high confidence.

In this paper, we aim to shed light on the internal mechanism of language models by studying their behavior on regular languages (Chomsky, 1959; Hopcroft et al., 2007). Regular languages, defined

Table 1: Summary of datasets/languages.

| Name | regex | #states. | #examples | examples | description | complexity | dependency |
|---|---|---|---|---|---|---|---|
| alter | $0(10)*$ | 3 | 44 | 0, 01010 | alternate 0 and 1 | $AC^0$ | local |
| mdY | $\backslash d\{2\}/\backslash d\{2\}/\backslash d\{4\}$ | 11 | 50000 | 09/12/2022 | real date strings of format m/d/Y | $AC^0$ | local |
| end0 | $(0|1)*0$ | 3 | 50000 | 110, 0010 | end with 0 | $TC^0$ | local |
| parity0 | $(1|01*0)*$ | 2 | 50000 | 1, 1010 | contain an even number of 0s | $TC^0$ | global |
| div3 | $(0|1(01*0)*1)*$ | 3 | 10000 | 00, 11, 1001 | binary integers divisible by 3 | $TC^0$ | global |

by specific regular expressions (regex) (Kleene, 1951), provide a controlled and well-understood framework for examining the learning and generalization capabilities of language models. Strings of a given regular language can be generated through a random walk on a finite state automaton (DFA), which is equivalent to the defining regex. Therefore, the regex or its equivalent automaton represent the compressed knowledge underlying the training instances. The central question we seek to address is:

> *Given a neural language model trained exclusively on strings conforming to a regular expression, can one recover an equivalent automaton from it?*

If successful, such recovery would provide insights into how language models compress and represent linguistic knowledge, and qualify the knowledge they have acquired.

We focus on two prominent architectures: LSTM (Hochreiter & Schmidhuber, 1997) and GPT (decoder-only transformers) (Vaswani et al., 2017; Radford et al., 2019). We propose a *hardening* process to convert a language model into an equivalent finite automaton, a method we term LaMFA (Language Model to Finite Automaton). Given a trained language model, we begin by sampling strings it generates. We then discretize the state space using clustering techniques such as $k$-means. For LSTMs, the state space is naturally defined as its latent space. For GPTs, we hypothesize that the latent space immediately preceding the final linear layer serves as the state space. Next, we merge and denoise the states using the estimated transition matrix based on the existing partition, thereby reducing potential redundancy. A DFA is then computed using the final state partition and the transition matrix. An equivalent regex can further be obtained using the state elimination method (Brzozowski & McCluskey, 1963).

We conducted experiments on five different regular languages, varying in their circuit complexity (Arora & Barak, 2009) and context dependency, as shown in Table 1. Our experiments reveal several key insights into the behavior of language models on regular languages. We find that all models perform exceptionally well on languages with local context dependency, regardless of circuit complexity. However, languages requiring global context pose significant challenges, especially for LSTM models. Notably, LaMFA successfully extract DFA from the trained models, which often demonstrate improved validity rates and strong generalization capabilities. In some cases, these extracted DFA achieve high consistency with the original one, while in others, they encode more states, particularly in larger models. These findings suggest a complex interplay between model architecture, size, and the nature of the language being modeled.

The contributions of this paper can be summarized as follows.

- We conduct experiments on five regular languages with varying complexity, to investigate how linguistic knowledge is encoded and compressed in language models.

- We propose a simple method, LaMFA, to recover the knowledge from trained language models and empirically show that it can effectively extract DFA of high consistency with the neural model;

- Our observations draw new insights of the complex interplay between model architectures, language complexity, and the structure of extracted DFA;

- We argue that this pipeline potentially serve as a benchmark for improved interpretability of language models. We release all codes as well as the checkpoints of language models in the experiments

## 2 RELATED WORK

Many efforts have been made on explaining the knowledge captured by the neural language model for safety or ethical concerns, and its further developing (Madsen et al., 2022). Given the complex nature of both natural language and deep networks, existing explanation methods are based on *knowledge probing*, i.e. inspect the existence of specific knowledge in the model through prediction tasks or ablations (Tenney et al., 2019; Dalvi et al., 2022; Jiang et al., 2020; Shi et al., 2016; Meng et al., 2023; Madsen et al., 2022; Allen-Zhu & Li, 2024b). Such probing is conducted at different levels. For example, Jiang et al. (2020) assess the storage of factual knowledge through automatic prompting. Other existing works use predicting tasks to probe the existence of specific types of linguistic information in the hidden layers (Shi et al., 2016; Tenney et al., 2019). Meng et al. (2023) identify neurons associated with specific factual knowledge by causal interventions. Although probing helps in locating knowledge, the overall generating mechanism of the language model remains unexplained.

Recent works focus on assessing the expressive power of neural networks with their ability to *recognize* formal languages. Theoretically, LSTMs have been demonstrated to be strictly more powerful than regular languages, capable of perfectly emulating finite-state automata Merrill (2019). Empirically, Gers & Schmidhuber (2001), Sennhauser & Berwick (2018) and Bhattamishra et al. (2020b) have evaluated the potential of LSTMs to acquire context-free grammars. Regarding transformers, theoretical limitations have derived for different restricted form of transformers on recognizing formal languages of different circuit complexity (Hahn, 2020; Hao et al., 2022; Merrill et al., 2022; Merrill & Sabharwal, 2023; Li et al., 2024). For example, Merrill & Sabharwal (2023) show that log-precision transformers Merrill & Sabharwal (2024) are upper-bounded by uniform $TC^0$, i.e. they are only possible to compute formal grammars that can be simulated by a circuit in uniform $TC^0$. Empirically, Bhattamishra et al. (2020a) examined LSTM and encoder-only transformers' ability to recognize regular languages and implement counter mechanisms. Liu et al. (2023) demonstrated that transformers can learn automata with fewer layers than theoretically expected.

The extraction of deterministic finite automata from RNNs that recognizing formal languages has been extensively studied over the past few decades. (Giles et al., 1991; Omlin & Giles, 1996; Das & Mozer, 1993; Weiss et al., 2018; Michalenko et al., 2019). Early work by Giles et al. (1991) and Omlin & Giles (1996) focus on simple second-order RNNs. More recently, (Weiss et al., 2018) extended this study to more complex architectures such as GRU and LSTM. Our work builds upon this foundation by further extending the extraction process to transformer-based models.

A key distinction of our study is its focus on generative probabilistic language models, whereas previous works primarily examined RNNs and transformers trained on language recognition tasks, which result in deterministic models. By investigating generative language models, our research complements and expands upon this established body of work. Concurrent work by Allen-Zhu & Li (2024a) aligns with this effort. They focus on a family of synthetic context-free languages exhibiting hierarchical structures. By probing the trained model's latent states quantify attention patterns, they suggest that GPT models learn CFGs by implementing a dynamic programming-like algorithm. In comparison, we focus on regular languages, which provide a simpler yet powerful framework for analyzing model behavior, allowing us to precisely control the complexity and context dependency of the input. By utilizing finite automata as our analytical tool, we can examine both RNN-based and transformer-based architectures through a unified lens.

## 3 PRELIMINARY

As we focus on training datasets where all examples are strings matching certain regular expressions, we briefly introduce two closely related and equivalent notions: regular expressions (regex) and deterministic finite automata (DFA).

**Regular languages and regular expressions.** Given an alphabet (sometimes also called a *vocabulary*), i.e. a finite set $V$ of characters (e.g. $V = \{0, 1\}$), let $V^*$ denote all *words* consisting of characters in $V$. A *language* $L$ is a subset of $V^*$, i.e. $L \subseteq V^*$. A *regular language* is a language that is recursively defined as one of the following cases: (1) $\emptyset$ or $\{c\}$, where $c \in V$; (2) $L_1 \cup L_2$; (3) $L_1 L_2 := \{w_1 w_2 | w_1 \in L_1, w_2 \in L_2\}$, i.e. concatenation; (4) $L_1^* := \{w_1 w_2 ... w_n | n \in \mathbb{N}, w_i \in L_1, i = 1, ..., n\}$, where $L_1$ and $L_2$ are regular languages. The

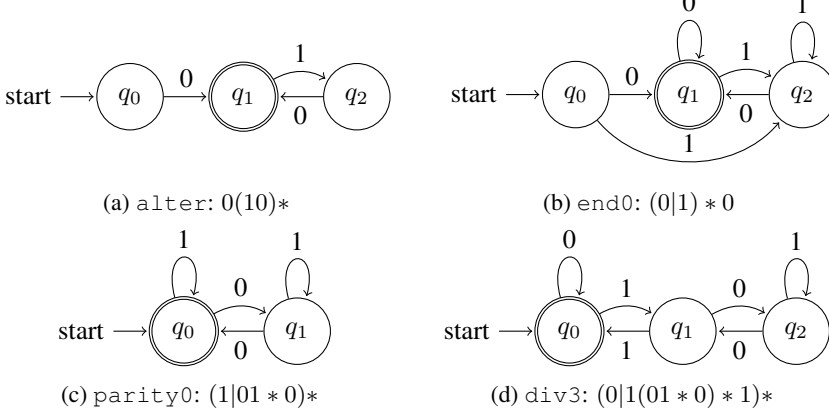

(a) `alter`: $0(10)*$

(b) `end0`: $(0|1)*0$

(c) `parity0`: $(1|01*0)*$

(d) `div3`: $(0|1(01*0)*1)*$

Figure 1: **Examples of deterministic finite automata (DFA) and their corresponding regular expression.** They are respectively DFA accepting strings that (a) `alter`: begin with 0 and followed by any number of copies of the string 10; (b) `end0`: end with 0; (c) `parity0`: contain an even number of 0s; (d) `div3`: are divisible by 3 when considered as an integer in base 2. Edges that do not point to any state are not shown.

unary operation '$*$' is called the Kleene star. A *regular expression* is a string specifying how a regular language is defined using the above recursive rules, and is recursively defined as one of the following cases: (1) $\epsilon$ (empty string) or $c$, where $c \in V$; (2) $r_1|r_2$ (or); (3) $r_1 r_2$ (concatenation); (4) $r_1*$ (Kleene star); where $r_1$ and $r_2$ are regular expressions. Common usage of brackets is also allowed.

**Deterministic finite automata.** A deterministic finite automaton can be considered as a special Turing machine where the machine can only read from left to right (i.e. one-way) and cannot write in the tape (i.e. read-only). Formally, a DFA is defined as a 5-tuple $(Q, V, \delta, q_0, F)$, consisting of (1) a finite set of states $Q$; (2) a finite set of input symbols called the alphabet (or vocabulary) $V$; (3) a transition function $\delta : Q \times V \to Q$; (4) an initial state (or start state) $q_0 \in Q$; (5) a set of accept states (or final states) $F \subseteq Q$ (often depicted with double circles). Some examples of DFA and regular expressions are shown in Figure 1. These DFA/regex are also used in our experiments.

**Equivalence of regular expressions and DFA.** Both regular expressions and DFA specify each a certain language $L \subseteq V^*$. It is a commonly known fact that regular expressions and DFA are equivalent in the sense that they both specify all regular languages. Algorithms exist for converting between regular expressions and DFAs, often utilizing non-deterministic finite automata (NFA) as an intermediate step (Kleene, 1956; McNaughton & Yamada, 1960). This equivalence allows us to use these representations interchangeably in formal language theory and practical applications.

## 4 METHODOLOGY

The pipeline of our proposed method LaMFA is shown in Figure 2. LaMFA begins with a trained language model. The language model is trained on a dataset of strings matching a given but unknown regular expression $r^*$ using an auto-regressive loss akin to GPT. Then, we sample strings $X_i$ using the language model (considered as a generative network) and do clustering on the features of all substrings $X_i[:t]$, i.e. the first $t$ characters of $X_i$, in the latent space before the last linear layer. Next, each substring $X_i[:t]$ is now attached to one center $c_{i,t} \in C$ of these clusters and we estimate a transition matrix $P \in [0,1]^{k \times |V| \times k}$ using all triplets

$$(c_{i,t}, X_i[t], c_{i,t+1}) \in C \times V \times C,$$

where $k := |C|$ is the number of clusters and $|V|$ is the alphabet size. To mitigate the effect of the randomness of the clustering algorithm and the noise, an additional merging and denoising procedure is applied to merge redundant cluster classes in $C$ and remove noisy transition patterns in $P$. A DFA is then obtained using the estimated transition matrix and a corresponding regular expression is computed using the classical *state elimination method* (Brzozowski & McCluskey, 1963). In the following, we give detailed introductions to the training settings and the LaMFA method.

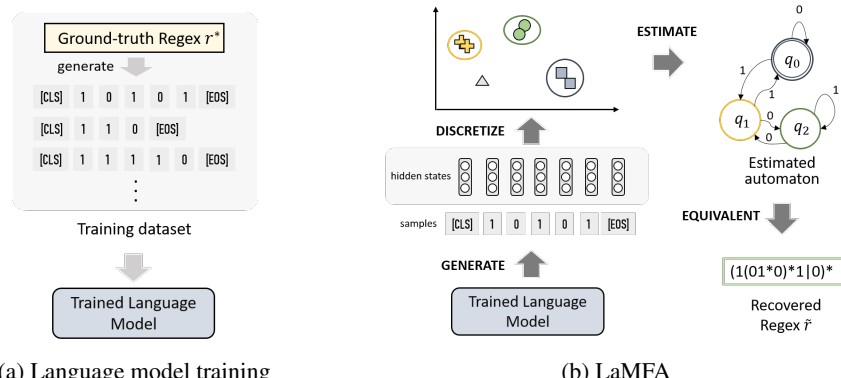

(a) Language model training          (b) LaMFA

Figure 2: (a) **Beforehand training.** A language model is trained on a dataset of strings matching an unknown regular expression. (b) **Pipeline of LaMFA**. Generate: strings are sampled using the trained language model. Discretize: the feature vectors before last linear layer of the substrings of sampled words are clustered in the latent space. Estimate: a deterministic finite automaton is computed via the estimation of the transition matrix. Finally the corresponding regular expression is obtained.

## 4.1 TRAINING DATA GENERATION

We select 5 simple regular languages to generate datasets for training the language model: `alter`, `mdY`, `end0`, `parity0`, and `div3`. The DFA of some are visualized in Figure 1. A summary of the datasets can be found in Table 1. Specifically, `alter` consists of strings that alternate 0 and 1, following the regex pattern $0(10)*$. `mdY` contains real date strings in the format mm/dd/yyyy, matching the regex $\backslash d\{2\}/\backslash d\{2\}/\backslash d\{4\}$. `end0` includes strings ending with 0. `parity0` contains strings with an even number of 0s. `div3` consists of binary integers divisible by 3.

These languages vary in their circuit complexity of their grammars. `alter` and `mdY` belong to the complexity group $AC^0$, i.e. they can be recognized by constant-depth circuit families with polynomial size (Arora & Barak, 2009). However the rest 3 languages are not, thus belong to the complexity group $TC^0$. According to previous empirical and theoretical studies (Bhattamishra et al., 2020a; Li et al., 2024), transformers struggles in recognizing regular languages outside $AC^0$. Thus it is interesting to examining if similar conclusion can draw in language generation ability.

Beyond circuit complexity, these languages exhibit varying degrees of context dependency. We define a language as having local context dependency if recognizing it requires only a constant-length context window. Conversely, languages with global context dependency necessitate information from the entire input sequence. Analysis of the regular expressions reveals that `alter`, `mdY`, and `end0` exhibit local dependency, whereas `parity0` and `div3` require global context. For example, `alter` can be recognized using a context window of merely two characters.

We consider a random walk on the DFA graph to generate strings. It starts from the initial state and terminating only on the final states. For each episode of the random walk, the characters on all traversed edges then form a valid string accepted by the DFA. The only randomness we need to introduce is in the choice of the next character to read. For this, we apply a uniform distribution on all possible actions/characters. Notably, there is an extra action 'terminate' in each final state. We follow this data-generating process to generate 10000 examples for `div3`, 50000 examples for `parity0` and `end0`. For `mdY`, we generate date strings from 01/01/1900 to 03/16/2023, in the m/d/Y format, with 50000 examples. For `alter`, we generate all 44 possible examples under the constraint on maximum length ($\leq 88$). Note that training data generated as above can contain repetitive strings.

## 4.2 KNOWLEDGE RECOVERING

In this subsection, we present the detailed process of recovering knowledge from a neural network. The algorithmic description of our method is illustrated in Algorithm 1 and Algorithm 2. Given a language model fully trained, we plan to recover the original knowledge from itself by hardening it. Specifically, LaMFA begins with generating a series of sequences by language models. For each

---

**Algorithm 1** LaMFA

---

**Input:** A trained language model $p_\theta$ with parameters $\theta$. From $p_\theta$ one also gets a function $\bar{p}_\theta : V^* \to \mathbb{R}^d$ that computes feature vectors before last layer.
**Input:** $N$: number of strings to sample.
**Input:** $K$: number of clusters.
**Output:** $P_{c,v,c'}$: transition matrix; $O_{c,v}$: output matrix.
1: $S \leftarrow \{X_i\}_{i=1}^N \sim p_\theta$    # Generate using LM
2: $S' \leftarrow \{X[:t]\}_{X \in S; t=1,\ldots,len(X)}$    # Consider all substrings of first $t$ characters
3: $H' \leftarrow \{\bar{p}_\theta(s')\}_{s' \in S'}$    # Compute feature vectors
4: $F \leftarrow KMeans(H', K)$, where $F : \mathbb{R}^d \to C$ and $C := \{1, \ldots, K\}$    # Discretize the feature vectors into clusters
5: $E \leftarrow \{(F(X[:t-1]), X[t], F(X[:t]))\}$, where $X \in S; t = 2, \ldots, len(X)$    # Construct triplets
6: $P_{c,v,c'} \leftarrow \#\{e \in E | e_1 = c, e_2 = v, e_3 = c'\}/\#\{e \in E | e_1 = c, e_2 = v\}$    # Estimate the transition matrix
7: $O_{c,v} \leftarrow \#\{e \in E | e_1 = c, e_2 = v)\}/\#\{e \in E | e_1 = c)\}$    # Estimate the output matrix
8: $P_{c,v,c'}, O_{c,v}, F \leftarrow Merge(P_{c,v,c'}, O_{c,v}, F, K)$    # See Algorithm 2
9: **return** $P_{c,v,c'}, O_{c,v}$.

---

**Algorithm 2** Merge

---

**Input:** $P_{c,v,c'}$: transition matrix, $O_{c,v}$: output matrix, $F$: clustering function
**Input:** $K$: number of clusters.
**Output:** $P_{c,v,c'}^*$: new transition matrix, $O_{c,v}^*$: new output matrix, $F^*$: new clustering function
1: $k \leftarrow K$
2: **repeat**
3:    $i, j \leftarrow \arg\max_{i,j} \text{cosine\_similarity}\left([\text{flatten}(P_{i,:,:}), O_i], [\text{flatten}(P_{j,:,:}), O_j]\right)$
   # Find the most two similar clusters $i, j$
4:    $\{F | C = i\} \leftarrow \{F | C = j\}$    # Merge cluster $i, j$
5:    Denoise operation shown in Equation (2).
6:    Update $P_{c,v,c'}, O_{c,v}, F$ for merged clusters.
7:    Update the best tuple $P_{c,v,c'}^*, O_{c,v}^*, F^*$ (according to valid rate).
8:    $k \leftarrow k - 1$
9: **until** $k = 1$
10: **return** $P_{c,v,c'}^*, O_{c,v}^*, F^*$.

---

generated sequence $X_i$, the hidden representation of each token $X_i[t]$ in the last Transformer/LSTM layer is extracted, noted by $h_i[t] \in \mathbb{R}^d$. $d$ is the hidden dimension of the language model. The hidden state encodes the substrings $X_i[:t]$ and ideally corresponds to the DFA states. We denoted all of the collected hidden states as set $H = \{h_i[t]\}$. Subsequently, LaMFA utilizes the $k$-means algorithm (Ahmed et al., 2020) to cluster the collected hidden states $h_i[t]$ into $k$ clusters. After clustering, the 5-tuple components of DFA can be obtained:

- $Q$: the finite set of states $Q$ is denoted as the clusters in $k$-means algorithms, which have $k$ different states.

- $V$: the input symbols set $V$ corresponds to the vocabulary of generated sequences.

- $\delta$: illustrated in Algorithm 1, by counting the number of transitions between two consecutive tokens, LaMFA can construct a transition matrix $P$ of dimensions $k \times |V| \times k$, where $P_{ijk}$ represents the frequency of transitions from cluster $i$ to cluster $k$, given the input $V_j$. After normalization, this leads to the formulation of the corresponding transition function $\delta$.

- $q_0$: the starting state $q_0$ is denoted as the cluster corresponding to the special token <bos>'s hidden state.

- $F$: the final states $F$ are correspond to the clusters which can generate special token <eos>.

Note that DFA can only take tokens as inputs and thus can not generate sequences directly. To make it generative, LaMFA maintains an additional $K \times V$ frequency matrix $O$ where each row

represents the output token distribution for its corresponding state. By normalizing the frequency matrix $\omega = \text{diag}\left(\frac{1}{\sum_{k=1}^n O_{ik}}\right) O$, it becomes an output probability matrix which enables the recovered DFA to generate sequences.

Due to the unpredictability of cluster numbers and potential noise, it is important to allow a sufficiently large $k$ in the $k$-means algorithm. To achieve this we incrementally test larger values and evaluate the resulting DFA's accuracy. This process continues until the accuracy improvement falls below a threshold $\tau = 0.1$. After this step, LaMFA merges redundant clusters and removes noisy transition patterns in $P$. This approach ensures a precise mapping of the model's hidden states, accounting for the clustering algorithm's randomness and possible imperfections in the language model's training.

**Merging and denoising.** The merging and denoising procedures are illustrated in Algorithm 2. The merging procedure aims to combine similar and redundant clusters. For each merge step, LaMFA greedy merges the two most similar clusters. Specifically, to find the most similar clusters, LaMFA first reshapes the transition matrix $\delta \in \mathbb{R}^{K \times |V| \times K}$ into $|V|$ individual matrices, each of size $\delta_{1..|V|} \in \mathbb{R}^{K \times K}$. Subsequently, we concatenate these $|V|$ matrices as well as normalized frequency matrix $\omega \in \mathbb{R}^{K \times |V|}$ along their second axis which forms the characteristic matrix $M \in \mathbb{R}^{K \times (|V| \times K + |V|)}$:

$$M = [\delta_1, \delta_2, ..., \delta_{|V|}, \omega]$$

where $[\cdot, \cdot]$ denotes the concatenation operation. Each row in the characteristic matrix $M$ depicts the corresponding cluster's outgoing transition behavior under all circumstances. Finally, the most similar two cluster is obtained by calculating the cluster-to-cluster similarity matrix $MM^T$ and picking out the cluster pair with the highest similarity score. LaMFA then re-calculates the new transition matrix $P$ and frequency matrix $O$ by treating these two clusters are one. Additionally, a denoising operation is performed on top of the newly obtained $P$ and $O$ before normalization. Specifically, sharpening is performed in all $|V|$ slice of $P_{:,v,:}$ (where $v$ ranges from 1 to $|V|$):

$$P'_{k,v,j} = \frac{P_{k,v,j}^{\frac{1}{T}}}{\sum_{l=1}^{K} P_{k,v,l}^{\frac{1}{T}}} \sum_{l=1}^{K} P_{k,v,l} \tag{2}$$

For frequency matrix $O$, we set all frequencies under threshold $\tau_o$ to zero to abandon the noisy frequency signal. After the denoising operation, we obtained new $\delta$ and $\omega$ by normalizing $P$ and $O$. Intuitively, removing the noisy pattern in the $\delta$ and $\omega$ will increase the resulting automaton's accuracy. LaMFA utilize this heuristic by greedily merging similar states until the resulting automaton's accuracy begins to decrease. Merging and denoising are iteratively conducted for $K$ steps.

After this step, a finite automaton, which is probably non-deterministic will be acquired. We convert it into a DFA with the classical subset construction algorithm (Rabin & Scott, 1959).

## 5 EXPERIMENTS

### 5.1 SETTINGS

**Dataset.** We experiment with the 5 datasets introduced in Section 4.1: `parity0`, `div3`, `alter`, `end0` and `mdY`. Each dataset is split into train/eval-ID/eval-OOD according to the ratio 3/1/1. The eval-OOD set is an out-of-distribution (OOD) evaluation set. For `parity0`, `div3`, `alter` and `end0`, eval-OOD sets consist of their longest 20% samples. For `mdy`, the eval-OOD set consists of date strings with the top 20% largest sum of digits. The eval-ID set and train set are random splits of the rest samples.

**Configuration and Evaluation.** The detailed architecture of the experimented language models is shown in Table 3. Details about the training hyperparameters are included in the Appendix. *Valid rate* (denoted as `valid` in tables) and the cross-entropy loss (denoted as `ce` in tables) are used as the evaluation metric for measuring the quality and diversity of the language model. To compute the valid rate, we generate 10,000 samples under each language model, and then test the validity of the generated sample by the ground-truth regex of the corresponding language. The valid rate is then defined as the ratio of the valid samples in all generated non-empty strings. The cross-entropy loss is calculated on the evaluation set to compare the relative distributional similarity between different language models and sample distributions.

Table 2: The valid rate and cross-entropy loss of different language models. "neural" denotes the raw trained models. "kmeans" denotes the model after the k-means step in LaMFA. "LaMFA-DFA" denotes the final hardened model. Three architectures are used: LSTM (0.50K), GPT-tiny (0.68K or 1.09K), and GPT-nano (86.16K). Underlined items correspond to cases where LaMFA recovers the exact/equivalent regular expression. Top values are bolded.

| Dataset | Model | neural | | kmeans | | | LaMFA-DFA | | |
|---------|-------|--------|------|--------|------|-----------|-----------|------|-----------|
| | | valid ↑ | ce ↓ | valid ↑ | ce ↓ | # cluster | valid ↑ | ce ↓ | # cluster |
| alter | LSTM | 98.33 | 3.80 | 90.46 | 3.84 | 10 | **99.33** | **3.78** | 3 |
| | GPT-tiny | 98.51 | 5.27 | 98.76 | 5.29 | 10 | **100.00** | **3.92** | 3 |
| | GPT-nano | 99.41 | 4.26 | 99.50 | 4.23 | 10 | **100.00** | **4.00** | 3 |
| mdY | LSTM | **92.91** | **10.97** | 86.62 | 11.6 | 72 | 90.00 | 11.66 | 41 |
| | GPT-tiny | **99.82** | **10.80** | 99.19 | 11.28 | 55 | **99.82** | 11.19 | 15 |
| | GPT-nano | **99.94** | **10.73** | 94.12 | 10.92 | 56 | 96.39 | 10.91 | 24 |
| end0 | LSTM | 99.92 | 4.12 | **100.00** | 4.12 | 5 | **100.00** | 4.12 | 3 |
| | GPT-tiny | 99.96 | **4.10** | 99.79 | 4.11 | 39 | **100.00** | 4.11 | 33 |
| | GPT-nano | 99.96 | **4.11** | **100.00** | 4.15 | 60 | **100.00** | 4.14 | 47 |
| parity0 | LSTM | **53.37** | 5.23 | 53.82 | 4.86 | 5 | **53.37** | **4.87** | 2 |
| | GPT-tiny | 70.40 | **4.39** | 70.29 | 4.44 | 65 | **70.44** | 4.43 | 45 |
| | GPT-nano | 98.05 | **4.02** | 98.31 | 4.48 | 74 | **100.00** | 4.67 | 3 |
| div3 | LSTM | 41.25 | 6.08 | 42.21 | 6.01 | 12 | **42.68** | **6.00** | 7 |
| | GPT-tiny | 56.07 | **5.68** | 56.40 | 5.72 | 39 | **57.17** | 5.72 | 34 |
| | GPT-nano | **85.43** | **5.28** | 84.42 | 5.46 | 96 | 85.05 | 5.42 | 77 |

Table 3: Architectures of language models. Time denotes the average time (ms) used for generating 10000 samples with GPU.

| Models | embed. dim. | layers | #param. | time (ms) |
|--------|-------------|--------|---------|-----------|
| LSTM | 6 | 1 | 0.50K | 38.7 |
| GPT-tiny | 6 | 1 | 0.68K | 2027.7 |
| GPT-nano | 48 | 3 | 86.16K | 11245.8 |

## 5.2 RESULTS

Our experiments yielded several significant insights into the behavior of language models when applied to regular languages.

### 5.2.1 LANGUAGE MODEL PERFORMANCE

The results in Table 2 reveal that the context dependency feature of regular languages has a more significant impact on language models' performance than circuit complexity. All models, regardless of their architecture, demonstrated exceptional performance on languages with local context dependency (i.e., alter, mdY, and end0), achieving an average accuracy of 98.34%. This high performance held true across various levels of circuit complexity.

However, languages requiring global context (parity0 and div3) presented significant challenges, particularly for LSTM models. While GPT-nano maintained relatively high performance with an average accuracy of 91.74% on these globally dependent languages, LSTM models showed a marked decrease in performance (53.37% on parity0, 41.25% on div3).

Among different architectures, GPT-nano consistently outperformed others, achieving valid rates higher than 98% across all five datasets. It's worth noting that neural language models, like most 'soft' algorithms, rarely achieve perfect (100%) accuracy.

Table 4: OOD performance of different language models on all datasets.

| Dataset | Model | neural | | LaMFA-DFA | |
|---|---|---|---|---|---|
| | | valid ↑ | ce ↓ | valid ↑ | ce ↓ |
| alter | LSTM | 96.26 | **5.14** | **99.75** | 5.27 |
| | GPT-tiny | 50.00 | 9.99 | **100.00** | **6.93** |
| | GPT-nano | 98.92 | 6.48 | **100.00** | **6.40** |
| mdY | LSTM | 93.15 | **10.85** | **98.06** | 11.11 |
| | GPT-tiny | 99.95 | **10.74** | **100.00** | 11.09 |
| | GPT-nano | 99.95 | 10.71 | **100.00** | **10.67** |
| end0 | LSTM | 99.84 | **13.57** | **100.00** | 13.61 |
| | GPT-tiny | 99.97 | **13.68** | **100.00** | 13.90 |
| | GPT-nano | 99.97 | **13.60** | **100.00** | 13.79 |
| parity0 | LSTM | **50.94** | 16.83 | 47.37 | **16.72** |
| | GPT-tiny | 50.52 | **21.89** | **52.02** | 32.15 |
| | GPT-nano | 64.52 | 47.12 | **100.00** | **21.19** |
| div3 | LSTM | **34.86** | 24.87 | 32.55 | **24.30** |
| | GPT-tiny | 30.70 | **30.59** | **32.56** | 41.59 |
| | GPT-nano | 32.11 | **31.44** | **36.66** | 42.09 |

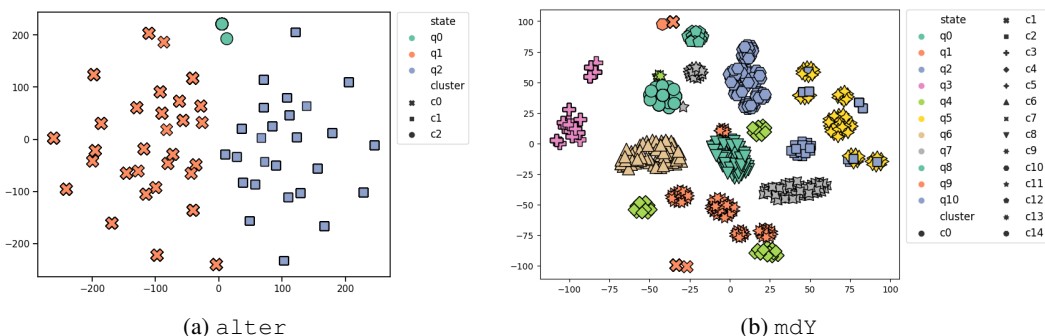

(a) `alter`                                          (b) `mdY`

Figure 3: The comparison between ground truth states and LaM clusters. The state shown in different colors denotes the ground truth state of DFA. The corresponding cluster shown in different shapes denotes the hardening result computed by LaMFA. (a) `alter` result computed using LaMFA recovering from GPT-nano. (b) `mdY` result computed using LaMFA recovering from GPT-tiny.

### 5.2.2 DFA EXTRACTION.

Comparing the neural and LaMFA-DFA column in Table 2, the extracted DFA by LaMFA show consistency with the original model in their validity and cross-entropy loss. To gain deeper insights, we visualized the states of the original generating DFA and the clusters defined by LaMFA. As shown in Figure 3 (a), the clusters are divided exactly the same as the original DFA states on `alter`. In Figure 3 (b), 15 clusters recovered from GPT-tiny on `mdY` also show highly consistent results.

In many cases, we observe that the number of states in LaMFA-DFA can be much larger than that of the ground truth minimal DFA, even when the validity is 100%. Figure 4 (a) visualize two LaMFA-DFA. It shows that LaMFA-DFA of LSTM on `end0` is exactly equivalent to the ground-truth regex. Interestingly, the DFA of GPT-tiny contains an extra state (0,1,2), which corresponds to the hidden refusing state for recognizing `alter` strings. These observations suggest that larger models may learn more nuanced representations of the language, potentially capturing subtleties beyond the minimal DFA representation.

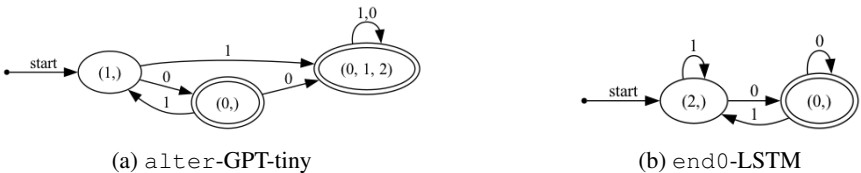

(a) `alter`-GPT-tiny  (b) `end0`-LSTM

Figure 4: The extracted DFA from (a) GPT-tiny on `alter`; (b) LSTM on `end0`.

### 5.2.3 GENERALIZATION CAPABILITY

LaMFA-DFA generally achieved better evaluation performance compared to the original neural models. For instance, recovered DFAs based on all three architectures reached 100% valid rate and lower cross-entropy on the `end0` dataset. Table 4 presents the out-of-distribution (OOD) generalization evaluation results. For GPT-tiny and GPT-nano, the hardened models consistently demonstrated higher OOD valid rates. For LSTM, on `alter`, `end0`, and `mdY`, LaMFA improved the valid rate while maintaining comparable cross-entropy loss.

Comparing OOD performance (Table 4) with in-distribution performance (Table 2), we noticed that both GPT-tiny and GPT-nano experienced significant drops in valid rate on `parity0` (from 70.4 to 50.52, and 98.05 to 76.56, respectively). Interestingly, these observation align with previous DFA extraction studies on language recognition RNNs (Giles et al., 1991; Das & Mozer, 1993), which showed that extracted rules often exhibit better generalization ability than the original neural models.

## 6 CONCLUSION AND DISCUSSIONS

This paper presents a pioneering study bridging probabilistic modeling with symbolic computation models (automata). By examining trained language models on regular languages of varying complexity, we demonstrate that context dependency is the dominant factor in language modeling complexity. This insight offers new perspectives on regular language complexity and the expressiveness of language models. Our proposed method, LaMFA, successfully extracted DFAs from trained models, often showing consistency with the original models in terms of validity and cross-entropy loss. In some cases, extracted DFAs captured more nuanced representations than the minimal ground truth DFA. LaMFA-extracted DFAs generally demonstrated better evaluation performance and improved out-of-distribution generalization compared to the original neural models, aligning with previous findings in DFA extraction studies. It marks a significant advancement in model interpretability and generalization. Our observations reveal a complex interplay between model size, language complexity, and the structure of extracted DFA.

This research complements existing work on regular language recognition models and opens new avenues for studying language models through the lens of symbolic computation. By establishing this connection, we pave the way for future investigations that combining probabilistic and symbolic approaches in computational linguistics and machine learning. Furthermore, we argue that this pipeline—training models on multiple different regular languages and investigating the extracted DFA—can potentially serve as a benchmark for analyzing language models of different architectures. To facilitate future development and research in this area, we are releasing all codes and checkpoints used in this study.

However, it's crucial to acknowledge the limitations of this study. First, our focus on formal languages, specifically regular languages, limits the direct generalization of our findings to natural language processing tasks, which involve far more complex linguistic structures and ambiguities. Second, while our DFA extraction algorithm yielded promising results, there is potential for developing stronger, more efficient algorithms that could extract even more accurate or compact automata representations. Finally, our experiments were conducted on relatively small-scale models compared to the massive language models currently at the forefront of AI research. Extending this work to larger-scale models could reveal different behaviors or challenges, particularly in terms of computational feasibility and the complexity of extracted automata. These limitations point to valuable directions for future research in this area.

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

# A  EXTENDED RELATED WORKS

**Probing knowledge in language models.**  Many efforts have been made on explaining the knowledge captured by the neural language model for safety or ethical concerns, and its further developing (Madsen et al., 2022). Given the complex nature of both natural language and deep networks, existing explanation methods are based on *knowledge probing*, i.e. inspect the existence of specific knowledge in the model through prediction tasks or ablations (Tenney et al., 2019; Dalvi et al., 2022; Jiang et al., 2020; Shi et al., 2016; Meng et al., 2023; Madsen et al., 2022; Allen-Zhu & Li, 2024b). Such probing is conducted at different levels. For example, Jiang et al. (2020) assess the storage of factual knowledge through automatic prompting. Other existing works use predicting tasks to probe the existence of specific types of linguistic information in the hidden layers (Shi et al., 2016; Tenney et al., 2019). Meng et al. (2023) identify neurons associated with specific factual knowledge by causal interventions. Although probing helps in locating knowledge, the overall generating mechanism of the language model remains unexplained.

**Symbolic regression.**  Symbolic regression is the task of learning a symbolic representation from *data*. For example, physics-informed neural networks (PINNs) (Raissi et al., 2019) aim at discovering the partial differential equations behind a given dataset. AI Feymann (Udrescu & Tegmark, 2020) also tries to rediscover equations in physics from data using neural networks. Different from symbolic regression, our method only relies on trained parameters and assumes no knowledge at all of the training data.

**Assessing neural networks with formal languages.**  Recent works focus on assessing the expressive power of neural networks with their ability to *recognize* formal languages. Theoretically, LSTMs have been demonstrated to be strictly more powerful than regular languages, capable of perfectly emulating finite-state automata Merrill (2019). Regarding transformers, theoretical limitations have derived for different restricted form of transformers on recognizing formal languages of different circuit complexity (Hahn, 2020; Hao et al., 2022; Merrill et al., 2022; Merrill & Sabharwal, 2023). For example, Hao et al. (2022) and Hahn (2020) have derived theoretical limitations for hard attention transformers, where attention distributions focus all probability mass on a single index. Their findings indicate that $AC^0$, the class of languages recognizable by constant-depth circuit families, serves as an upper bound for the formal languages that hard-attention transformers can recognize. Notably, the formal language `parity0` falls outside $AC^0$. Merrill & Sabharwal (2023) show that log-precision transformers Merrill & Sabharwal (2024) are upper-bounded by uniform $TC^0$, i.e. they are only possible to compute formal grammars that can be simulated by a circuit in uniform $TC^0$. Empirically, Bhattamishra et al. (2020a) examined Transformers' ability to recognize regular languages and implement counter mechanisms. Liu et al. (2023) demonstrated that transformers can learn automata with fewer layers than theoretically expected. Sennhauser & Berwick (2018) and Bhattamishra et al. (2020b) have evaluated the potential of LSTMs to acquire context-free grammars.

In this work, we focus on probabilistic language models, i.e. neural networks trained with the language modeling task, instead of recognition. Concurrent work by Allen-Zhu & Li (2024a) aligns with this effort. They focus on a family of synthetic context-free languages exhibiting hierarchical structures. By probing the trained model's latent states quantify attention patterns, they suggest that GPT models learn CFGs by implementing a dynamic programming-like algorithm. In comparison, we focus on regular languages, which provide a simpler yet powerful framework for analyzing model behavior. This approach allows us to precisely control the complexity and context dependency of the input. By utilizing finite automata as our analytical tool, we can examine both RNN-based and transformer-based architectures through a unified lens. Furthermore, this approach enables us to build upon previous theoretical works, highlighting the crucial distinctions between language generation and recognition tasks.

**Finite automata extraction.**  The extraction of deterministic finite automata from RNNs has been extensively studied over the past few decades. (Giles et al., 1991; Omlin & Giles, 1996; Das & Mozer, 1993; Weiss et al., 2018; Michalenko et al., 2019). Early work by Giles et al. (1991) and Omlin & Giles (1996) focus on simple second-order RNNs. Giles et al. (1991) pioneered this field by developing a dynamic clustering algorithm to extract production rules from trained second-order RNNs. This method involved state clustering, transition mapping, and graph reduction to obtain minimal DFA representations. They proposed that in some cases this approach often resulted in

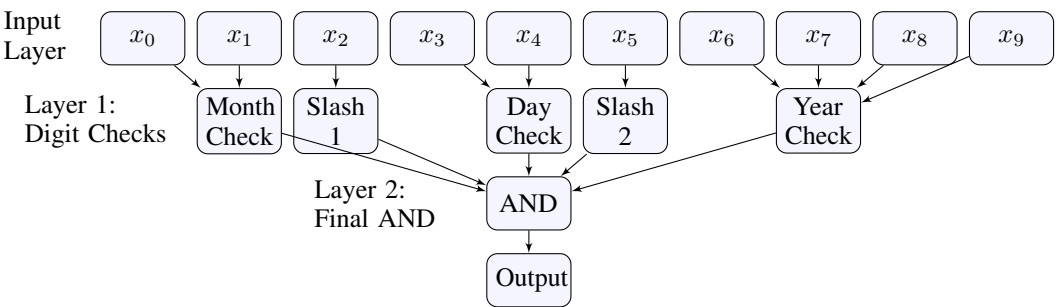

Figure 5: AC$^0$ Circuit for recognizing `mdY`.

Table 5: The hyperparameters

| Dataset | parity0 | div3 | alter | end0 | mdY |
|---|---|---|---|---|---|
| max length | 88 | 88 | 88 | 88 | 12 |
| batch size | 64 | 64 | 10 | 64 | 64 |
| optimizer | | | SGD | | |
| scheduler | | | CosineAnnealingLR | | |
| learning rate | | | 1e-3 | | |
| epoch | | | 500 | | |

extracted grammars that outperformed the original neural networks in classifying unseen strings. Omlin & Giles (1996) further introduce techniques to extract multiple consistent DFAs from a single network. They focused on improving rule quality and developed heuristics for selecting the most accurate DFA representation of the learned grammar. More recently, (Weiss et al., 2018) propose a new method using Angluin's $L^*$ algorithm with the trained RNN as an oracle to extract a DFA representing its behavior. They efficiently extracted accurate automata from complex networks, including GRU and LSTM architectures. By applying this technique to RNNs trained to 100% train and test accuracy on simple languages, they discover that some RNNs have not generalized to the intended concept.

Our work builds upon this foundation by further extending the extraction process to transformer-based models. A key distinction of our study is its focus on generative language models, whereas previous works primarily examined RNNs trained on language recognition tasks, which result in deterministic models. By investigating generative language models, our research complements and expands upon this established body of work.

## B   REGULAR LANGUAGES AND CIRCUIT COMPLEXITY

`alter` can be recognized by the AC$^0$ circuit because the language requires only local, fixed-distance checks that can be performed in parallel. The circuit uses a NOT gate to ensure the string starts with 0, followed by a layer of AND gates that check for alternating 1s and 0s in adjacent positions. These AND gates operate independently on different parts of the input, allowing simultaneous evaluation. A final OR gate combines these results. This structure maintains the key properties of AC$^0$: constant depth (three layers including input), polynomial size (linear growth with input length), and unbounded fan-in (at the OR gate). `mdY` can also be recognized with AC$^0$ circuit since it has fixed length and finite alphabet. We illustrate a feasible circuit in Figure 5.

## C   HYPER-PARAMETERS

The detailed hyper-parameters of experiments are illustrated in Table 5.

As the alphabets are simple, there is no need for tokenization and each character is considered as an independent token.

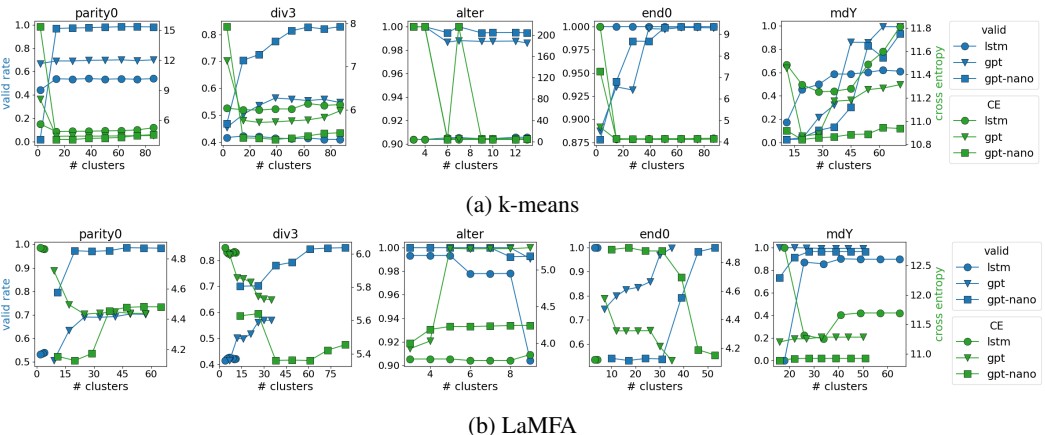

(a) k-means

(b) LaMFA

Figure 6: Valid rate and cross-entropy loss v.s. the number of clusters in k-means and LaMFA, respectively. (a) The k-means performance increases with the number of clusters. (b) LaMFA increases the generalization ability by merging the state from large clusters to ground truth clusters.

# D    MORE RESULTS

## D.1    CLUSTERING

We perform a study on the impact of the initial number of clusters as shown in Figure 6. Figure 6a illustrates how $k$ influences the method only with $k$-means clustering, and Figure 6b demonstrates how $k$ influences the whole algorithm LaMFA. As can be seen, larger $k$ usually has better performance, while it may lead to overfitting (with large cross-entropy) as it is the case for `mdY`, `div3` and `parity0`.

We show more clustering results for the datasets `div3` and `end0` in Figure 7. We can see that the estimated states (i.e. the clusters) for `end0` correspond well to the ground truth. But it is not the case for `div3`. This is due to the difficulty of the dataset `div3`, especially the fact that the language model over-fits the dataset and get almost random OOD generalization performance. This can be seen from the results in Tables 2 and 4.

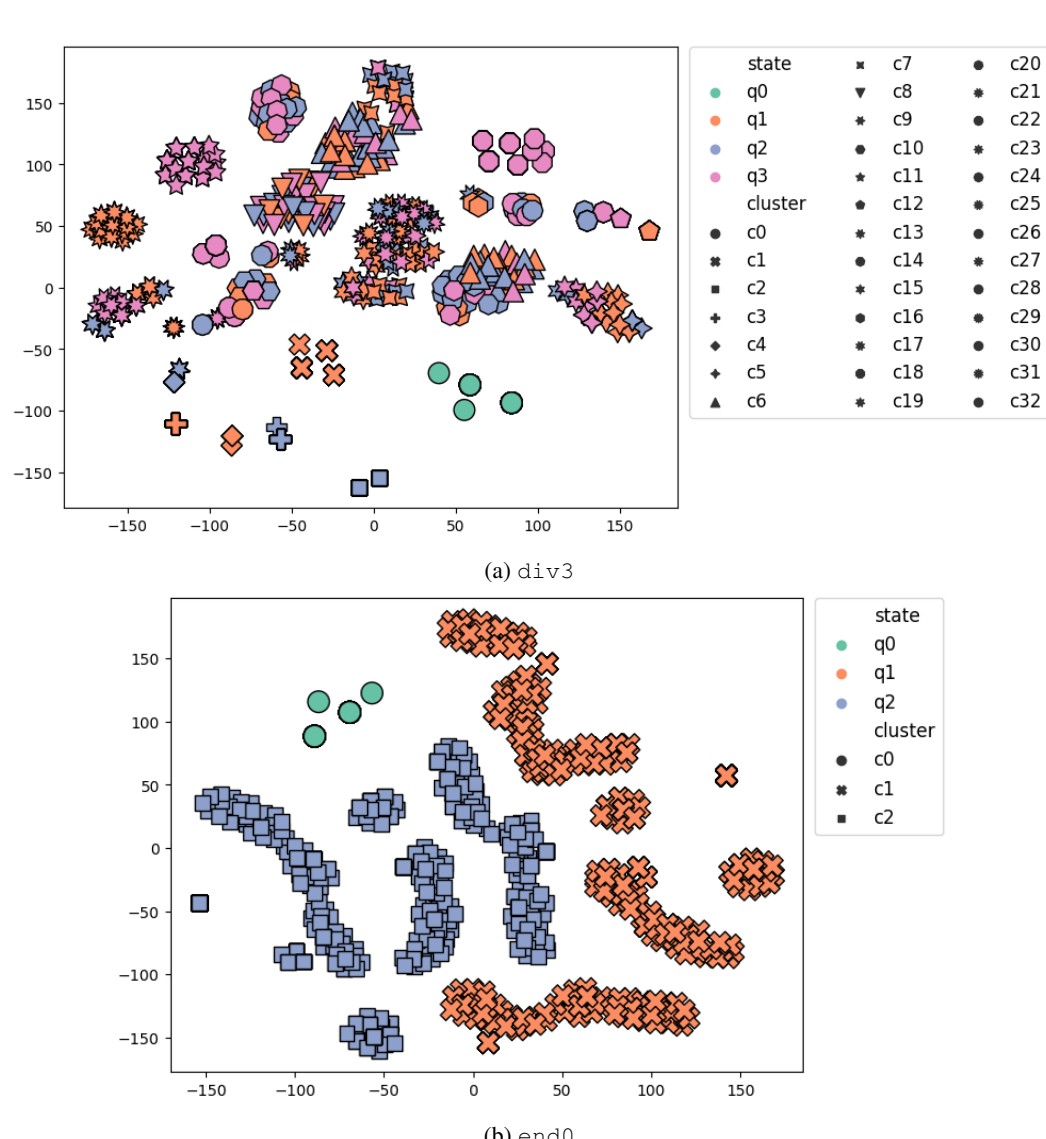

Figure 7: The comparison between ground truth states and LaMFA clusters, as a continuation of Figure 3. (a) `div3` result computed using LaMFA recovering from GPT-nano. (b) `end0`, using LSTM. We can see that the estimated states (i.e. the clusters) for `end0` correspond well to the ground truth. But it is not the case for `div3`, due to the difficulty of the dataset.

