# OpenReview forum: "Recovering Knowledge by Hardening Language Models"
_ICLR.cc/2025/Conference — ICLR 2025 Conference Withdrawn Submission_

### Official Review · Reviewer_g6po · 2024-10-29

**Soundness:** 2
**Presentation:** 2
**Contribution:** 2
**Rating:** 3
**Confidence:** 4

**Summary:**

The authors propose a method of converting an LSTM or GPT LM trained on a regular language to a deterministic finite automaton (DFA) that approximates the LM. They test their method on 5 regular languages belonging to two different circuit complexity classes (AC0 and TC0). The languages also have different "context complexity", in that some only require a local, finite suffix of a string to correctly recognize, and some require global context. They define a regex/DFA for each language and use it to sample datasets of positive examples. They train neural LMs on it from scratch. Afterwards, they extract the hidden representations for each prefix of each string in the dataset. They apply k-means clustering and denoising to convert them to states of a DFA. They find that the extracted DFA often has higher precision than the original LM, and that neural LM accuracy depends mostly on context complexity.

**Strengths:**

This paper proposes a method for extracting DFAs from neural LMs that have only been trained on positive examples and does not require negative sampling. To my knowledge this has not been tested on transformers before.

**Weaknesses:**

1. The number of languages (5) is small, and they are quite simple. It is not clear if this method scales up to more complicated languages where the minimal DFA has more than 3 states.
1. There are issues with the mdY language. It seems like the authors only allow semantically valid dates (i.e., they do not allow strings like 99/99/9999). This means the actual language tested, which incorporates these constraints, is different from the regex they report in the paper and requires a lot more than 11 DFA states. Also, when splitting examples into training and test data, there are constraints placed on the training data based on the sum of the numbers in the string, and this makes it more difficult to characterize the language actually represented in the training data. It should be noted that this language contains only a finite number of strings.
1. A technical comparison against prior work on converting neural LMs to DFAs, particularly Weiss et al. (2018), is conspicuously absent. What advantage does your method have vs. theirs? How is it different? The state merging algorithm seems to be very similar to the standard DFA minimization algorithm.
1. I think the alter language has been misclassified as having local instead of global context dependency.
1. Some aspects of the method are unclear. In particular, how do you determine when you have converged on the minimal number $k$ of states? I do not see a stopping condition for merging states in the pseudocode. How do you pick the sharpening parameter $T$ and the frequency threshold $\tau_0$?
1. There are issues with the evaluation metrics. For one, the accuracy metric only measures precision, not recall. For example, it is possible for an LM to achieve 100% accuracy by assigning all probability mass to a single string in the language. Including cross-entropy does mitigate this because it measures recall, but the authors do not provide what the lower-bound cross-entropy would be under the original DFA LM, so it is not clear when the models are doing well.
1. The authors claim that context dependency matters more than complexity class, but there is only one language (end0) where these two do not coincide. The paper would benefit from additional languages where they do not coincide. As mentioned above, alter is actually global, not local.
1. There are grammatical issues throughout the paper which affect its clarity, and it would benefit from a round of proof-reading.

**Questions:**

1. What would you say is the primary goal of the paper? Is it about interpretability of neural LMs, or improving the performance of neural LMs on regular languages by converting them to DFAs? If you start with a DFA to generate your data, what is the point of training a neural LM and converting it back to a DFA?
1. 485: Related to the above -- What do you think these subtleties might be? In this paper, most languages can be described by a DFA with $\leq 3$ states, so there are no extra subtleties to learn.
1. If interpretability is the primary goal, how would you scale this up to more complex languages? Code and natural language are not regular, so converting a LM for code or natural language to a DFA is unlikely to be informative.
1. For the language mdY, do you only allow valid dates, or are strings like 99/99/9999 allowed?
1. What is the purpose of using both DFAs and regexes, and not just one of the two? Why is it necessary to define the language in terms of a regex and convert the final DFA back to regex, if the data sampling is done using a uniform-weighted DFA and the model is converted to a DFA?
1. Can you give a formal definition of context dependency? I think the alter language is misclassified as local instead of global.
1. What is the longest length in the training set after splitting?
1. 340: I'm a bit confused by the formula for $M$. Why is $\omega$ included? The dimensions for $\delta_i$ and $\omega$ don't seem to match on the second axis, so I'm not sure how concatenation is possible. What would it mean for $\delta_i$ to be merged with a row of $\omega$? One is a distribution over destination states, but the other is a distribution over symbols.
1. 323: How do you get the frequency matrix $O$? You can get non-zero counts out of the same source state to multiple destination states on the same symbol. Do you marginalize over destination states?
1. Table 3: How do you measure ce for LaMFA-DFA?
1. Alg 2, line 4: What does this notation mean?
1. Alg 2, line 9: Do you always end with $k = 1$ cluster?
1. Fig 3b: Can you provide a DFA for mdY? Otherwise, I'm not sure how to interpret this.
1. With regard to the concept of local context complexity, this paper about n-gram LMs and transformers is relevant: https://arxiv.org/abs/2404.14994

Minor comments:
1. Parity is conventionally defined to be the language of strings with an odd number of 1s.
1. 141: Training NNs on language recognition (as opposed to language modeling) does not result in deterministic models per se -- what do you mean by this?
1. In Fig 1a, there is a missing transition going out of q0 on 1 (your definition of DFA requires a complete transition function).
1. 263: What do you mean by repetitive strings?
1. 376: Why do you eliminate empty strings?

---

### Official Review · Reviewer_nkYT · 2024-10-29

**Soundness:** 1
**Presentation:** 2
**Contribution:** 1
**Rating:** 1
**Confidence:** 5

**Summary:**

**TL;DR:** A deeply misguided, inconsistent, and confused text further burdened by its poor comprehension of the subfield's literature and self-congratulatory overselling.

The paper presents LaMFA, a method for constructing finite automata from hidden states of language models. The design decisions comprising LaMFA outnumber the automata samples/experimental setups (a grand total of 5) by a factor of 2, thus making the first doubts of the interpretability of the results. The method presented, LaMFA, is an apparent simplification of the Louvain method heavily tailored toward the carefully chosen experimental automata.

Somewhat grandiosely, he paper claims to "shed light on the internal mechanism of language models" (L053), "investigate how linguistic knowledge is encoded and compressed" (L098), "draw new insights on the interplay between architectures and language complexity" (L103), "yield significant insights into the behavior of LMs" (L415), and "mark a significant advancement in model interpretability and generalization" (L521). None of these claims about the contents of the paper are true, as most of these have no experimental counterpart in the text, and virtually every conclusion about the nature of language models that is drawn can be traced back to a design decision.

As a brief example of the prevalence of these issues: A claim is made that "the LaMFA-DFA of LSTM is exactly equivalent to the ground-truth regex" (L482). However, this is only a consequence of Algorithm 2 merging the previously disjoint cluster states. Okay, a counter-claim can be made that this is the intended function of Algorithm 2. But why then does "the DFA of GPT-tiny contain an extra state" (L483) even after the Algorithm 2 merge that was supposed to collapse it? You can't have it both ways; the successful recovery and unsuccessful recovery of a DFA cannot both be findings simultaneously. This would perhaps be acceptable if we had more samples and could draw statistical conclusions, but 5 languages are not nearly enough to do so -- a statistically significant sample should be considered instead and evaluated in an automated fashion (see Weakness 7 for more guidance on this). By the way, the DFA alter-GPT-tiny is incorrect as it also accepts the string "1" which is not in the alter language. All in all, the immediately-following claim that "these observations suggest that larger models may learn more *nuanced* representations of language" is a deliberate obfuscation of the contradictions arising in L480-L485. The use of *nuanced* is nothing less than a disconcerting, blatant attempt to pass "it worked once and it did not work once" for a finding.

While I touch on more issues with the paper in the weaknesses, it is important to note that this work is also flawed *conceptually*. In a surprising omission of related work, the text appears to be utterly oblivious to the results of H. Siegelmann from the late 90s giving constructive proofs to the Turing-completeness of RNN-based architectures including GRU/LSTM (rather surprisingly, however, the text goes into great lengths to cite the likes of Kleene, Rabin, Chomsky, or McNaughton&Yamada). Many of the (non-)findings of Sections 5.2.1-5.2.2 are focusing on the artifacts of arguably improper gradient-based training schedule of the selected architectures, as each of the 5 regular languages considered can be realized as RNNs following the construction of the seminal proof. Since both LSTMs (dated work, Siegelmann) and Transformers (contemporary work, Svete) can be constructed by hand to recognize simple finite automata, the insights the paper sets out to provide are already familiar from theory, and its contribution is reduced to a tedious construction and an entirely unconvincing experimentation. See also Weakness 6.

It is my recommendation that the work is not published. It should be instead revisited from the very beginning, refocused on the contribution of LaMFA, supported by further experimentation vouching for its utility and advantages over the omitted previous work on state merging, and only then submitted for review.

**Strengths:**

None.

**Weaknesses:**

1. The design decisions described in the paper outnumber the experimental data samples by a factor of 2 at the very least.
2. The paper refers to a considerable amount of obscure and/or dated work with little bearing on its goals.
3. An unsavory resemblance can be seen between this work and past works of William Merrill.
4. Early sections of the paper are riddled with typos and grammatical mistakes, especially Section 2 (e.g. "further developing" -> "further development" L111, "inspect" -> "inspecting" L113, "they are only possible to compute formal grammars" -> this is not a sentence, you need passive L129, "knowledge recovering" -> "knowledge recovery" L265, etc.).
5. That this work is "investigating generative language models" is a bit of an overstatement as the DFAs recovered by LaMFA are ultimately classifying recognizers rather than generative LMs.
6. The dependence of Algorithms 1-2 on the number and desired number of clusters (L274,L291) ultimately voids all claims about the complexity of the languages learned by the neural models. By varying the two Ks and given different orderings for the similarities of the clusters, one can arrive at different DFAs. Ultimately, we know that equivalent DFA constructions exist, so any claim on the complexity or imperfection of recovery based on the outputs of the method can be  attributed to the gradient-based training of the neural model as well as the proposed LaMFA method itself.
7. To give credibility to the claims about the effectiveness of LaMFA, the work should consider a population of automatically generated small DFAs/regular languages, not just 5 handpicked examples. For example, to have 90% confidence that the reconstruction success rate lies between 45 and 55%, at least 273 distinct DFAs should be considered.

**Questions:**

1. Why wasn't "Extracting Finite Automata from RNNs Using State Merging" (Merril, 2022), with clear (and somewhat unnerving) similarity to this work, cited? It is obviously previous art to this effort. Why was no comparison made in Section 2 "Related Work"?

---

### Official Review · Reviewer_Pu9Q · 2024-11-01

**Soundness:** 2
**Presentation:** 3
**Contribution:** 1
**Rating:** 3
**Confidence:** 4

**Summary:**

This paper introduces a method for extracting finite-state automata (FSAs) from trained language models, called LaMFA. The approach clusters string prefix representations in the model’s representation space and builds FSAs based on these clusters. The authors add a new denoising step to make clustering more stable and accurate. They apply LaMFA to both transformer and LSTM language models trained on five regular languages and compare the extracted FSAs to the actual FSAs for these languages. They find that the extracted automata are usually not identical to the original ones—they’re often larger—but they tend to generalize better than the models from which they were extracted.

**Strengths:**

This paper contributes to an important area of research: analyzing black-box models using interpretable, easier-to-analyze representations. Key strengths for me include:
- Exploring language models with interpretable structures like FSAs can lead to better understanding and analysis of these models, which is valuable for the field.
- The FSA denoising procedure seems new and intuitively useful, making the clustering process more robust.

**Weaknesses:**

There are areas where the paper could be improved:

- **Novelty in Methodology**: Apart from the new denoising step (the benefit of which isn’t explicitly measured), the overall approach seems similar to past work, especially to methods by Weiss et al. (2018), which—contrary to the claims in the paper—also work on *any* language model and come with some theoretical guarantees. The paper would benefit from a closer comparison to Weiss et al., as well as other FSA extraction methods, such as those by Merrill et al (2022) and other authors referenced in the paper.
- **Related Work Section**: The extended related work in the appendix mostly repeats what is in the main text. Instead, it would be better to replace this with a direct comparison to specific existing FSA techniques.
- **Limited Empirical Basis**: One main takeaway of the paper is that local context dependency is a better predictor of transformer performance than concepts like circuit complexity. However, since the models are trained on only five very simple languages, this evidence may not be enough to support that claim. Testing on more complex—and many more—languages would strengthen this finding.
- **Lack of Novelty in Reconciling Generative Models with Recognizers**: The reconciliation of LMs as generative models with FSAs as recognizers is useful, but the approach here is not the first to do so. Weiss et al. (2018), for example, presented an algorithm that addresses similar scenarios.
- **Valid Rate Measure**: The paper uses “valid rate” to evaluate how many generated strings are syntactically correct, implying this tests language diversity. But as defined, it seems to only measure precision—if a model only learned one syntactically correct string, it could achieve a perfect valid rate. Adding a diversity measure or clarifying this metric could provide a fuller picture.

**Questions:**

- Did you perform any quantitative tests on the effects of cluster denoising? How does it impact FSA extraction and the performance of the final FSA?
- What is the connection between your use of local context dependency and strictly local languages (or n-gram models)? There is quite a bit of literature on analyzing transformers with n-gram models, which might relate to your findings.
- How many models were trained on each dataset? How stable are the results across different training runs?
- A minor point: It seems the end0 language could be recognized by a two-state FSA, where the second state is reached on reading a zero.
- I’m not sure about the utility of Appendix B. It’s not mentioned in the main text, and while it makes a formal claim, the support is limited. It also suggests that alter can be recognized by AC0 due to local dependencies. However, AC0 circuits can handle long-distance dependencies too, so this reasoning might need clarification.
    - On a similar note, if the paper discusses circuit complexity and its relationship to FSA learnability, a brief introduction to circuits might help, similar to the background provided for FSAs.

---

### Official Review · Reviewer_Pt5K · 2024-11-04

**Soundness:** 3
**Presentation:** 3
**Contribution:** 2
**Rating:** 5
**Confidence:** 3

**Summary:**

This paper studies how language models learn regular languages. It proposes an algorithm LaMFA that can recover the finite state automaton corresponding to the regular language from the neural language model trained on the same language.

LaMFA first constructs the states in DFA by clustering the features generated in language models and then estimates the transition function by looking at strings with and without the last token.

They found that LaMFA-extracted DFAs can recover the full knowledge of the regular language and perform better than neural language models in out-of-distribution scenarios.

**Strengths:**

1. The method and results are neat, showing that neural language models are capable of learning the underlying DFA of a regular language from the strings of the language.

**Weaknesses:**

1. The findings about "extracted DFAs exhibiting more generalizability" kind of contradict the findings in [1], where they showed that a standard transformer often outperforms a transformer that's constrained to be compilable to a RASP program [2].
2. The results on regular languages are great, but how are they helpful for explaining how and why real-world language models work?
3. The results from [3] seem more generalizable and (slightly) earlier to me.

### Reference

[1] Learning Transformer Programs, https://openreview.net/forum?id=Pe9WxkN8Ff
[2] Thinking like transformers, https://proceedings.mlr.press/v139/weiss21a/weiss21a.pdf
[3] Physics of Language Models: Part 1, Learning Hierarchical Language Structures, https://arxiv.org/abs/2305.13673

**Questions:**

Does the same method explain how language models learn semantics instead of syntax?

---

### Note · Authors · 2024-11-26

**Comment:**

Dear Reviewers,

We would like to express our sincere gratitude for the time and effort you have dedicated to reviewing our submission. After careful consideration, we have decided to withdraw our paper from the review process.

We agree that this work is not suitable for ICLR in its current stage and greatly value your insightful feedback. We will revise the paper according to your suggestions to improve its quality. Thank you for your efforts in making this research area better through your thoughtful reviews.

Best regards, Submission14201 Authors

**Withdrawal Confirmation:**

I have read and agree with the venue's withdrawal policy on behalf of myself and my co-authors.